# Spatial Gap-Filling of ESA CCI Satellite-Derived Soil Moisture Based on Geostatistical Techniques and Multiple Regression

**Ricardo M. Llamas** [1] , **Mario Guevara** [1] , **Danny Rorabaugh** [2] , **Michela Taufer** [2] and **Rodrigo Vargas** [1,*]

1  Department of Plant and Soil Sciences, University of Delaware, Newark, DE 19716, USA; rllamas@udel.edu (R.M.L.); mguevara@udel.edu (M.G.)

2  Department of Electrical Engineering and Computer Science, University of Tennessee, Knoxville, TN 37996, USA; dror@utk.edu (D.R.); taufer@utk.edu (M.T.)

*  Correspondence: rvargas@udel.edu; Tel.: +1-302-831-1386

**Abstract:** Soil moisture plays a key role in the Earth's water and carbon cycles, but acquisition of continuous (i.e., gap-free) soil moisture measurements across large regions is a challenging task due to limitations of currently available point measurements. Satellites offer critical information for soil moisture over large areas on a regular basis (e.g., European Space Agency Climate Change Initiative (ESA CCI), National Aeronautics and Space Administration Soil Moisture Active Passive (NASA SMAP)); however, there are regions where satellite-derived soil moisture cannot be estimated because of certain conditions such as high canopy density, frozen soil, or extremely dry soil. We compared and tested three approaches, ordinary kriging (OK), regression kriging (RK), and generalized linear models (GLMs), to model soil moisture and fill spatial data gaps from the ESA CCI product version 4.5 from January 2000 to September 2012, over a region of 465,777 km$^2$ across the Midwest of the USA. We tested our proposed methods to fill gaps in the original ESA CCI product and two data subsets, removing 25% and 50% of the initially available valid pixels. We found a significant correlation (r = 0.558, RMSE = 0.069 m$^3$m$^{-3}$) between the original satellite-derived soil moisture product with ground-truth data from the North American Soil Moisture Database (NASMD). Predicted soil moisture using OK also had significant correlation with NASMD data when using 100% (r = 0.579, RMSE = 0.067 m$^3$m$^{-3}$), 75% (r = 0.575, RMSE = 0.067 m$^3$m$^{-3}$), and 50% (r = 0.569, RMSE = 0.067 m$^3$m$^{-3}$) of available valid pixels for each month of the study period. RK showed comparable values to OK when using different percentages of available valid pixels, 100% (r = 0.582, RMSE = 0.067 m$^3$m$^{-3}$), 75% (r = 0.582, RMSE = 0.067 m$^3$m$^{-3}$), and 50% (r = 0.571, RMSE = 0.067 m$^3$m$^{-3}$). GLM had slightly lower correlation with NASMD data (average r = 0.475, RMSE = 0.070 m$^3$m$^{-3}$) when using the same subsets of available data (i.e., 100%, 75%, 50%). Our results provide support for using geostatistical approaches (OK and RK) as alternative techniques to gap-fill missing spatial values of satellite-derived soil moisture.

**Keywords:** soil moisture; remote sensing; geostatistics; gap-filling; mesonet

## 1. Introduction

Addressing global environmental challenges requires knowledge and information derived from the most accurate and complete available datasets. Soil moisture has an important role in the water and energy cycles and is regarded as one of the essential terrestrial climate variables [1] due to its influence on soil and atmosphere feedbacks. Furthermore, soil moisture is a critical input variable for applications such as climate modeling [2–4], agricultural planning [5,6], and carbon budget analyses [7,8]. Because

of the importance of soil moisture, there are many in situ monitoring networks, organized at the global [9], regional [10,11], or national-scale [12–15]. Despite these national to global efforts, there is still a challenge to represent spatially explicit soil moisture information across large regions related to spatial limitations of in situ ground measurements.

Soil moisture can be estimated using remote sensors (e.g., spaceborne radiometers and radar sensors) to provide coarse-scale estimates on a regular basis [9,16]. Examples of remote sensing soil moisture monitoring systems include NASA's Soil Moisture Active Passive (SMAP) [16], ESA's Soil Moisture and Ocean Salinity (SMOS) [17], and the European Space Agency Climate Change Initiative (ESA CCI) [11,18] that deliver publicly available data for a wide range of applications. Despite advances in remote sensing technology, there are still large areas where soil moisture information is not regularly acquired, yielding information gaps in time and space across the world. Missing information arises from certain circumstances such as high canopy density, snow and ice cover, extremely dry surface conditions, or frozen soil [11]. These factors hinder radiometers or radar sensors in measuring the dielectric constant in the top layer of soil in order to estimate the water content [19].

Consequently, there is a need to develop gap-filling strategies to provide spatially complete satellite-derived soil moisture data across the world. In the most recent version of the ESA CCI product (version 4.5), soil moisture values are derived from the combination of active and passive sensors based on a weighted mean, proportional to the signal-to-noise ratios (SNRs) [20]. In areas where soil moisture information cannot be derived using SNRs, values are estimated using a polynomial regression between the signal-to-noise ratios [20]. Version 4.5 masks areas of dense vegetation using vegetation optical depth layers and flags measurements under frozen conditions [21]; consequently the product has multiple gaps across the world [22].

Other statistical methods (e.g., discrete cosine transformations and singular spectrum analysis) have been applied to fill spatial gaps for satellite-derived geophysical datasets, as well as soil moisture from field measurements [23–25]. These approaches are focused either on the statistical distribution of the data or three-dimension information, which includes both space and time. We postulate that alternative gap-filling methods could take advantage of the information contained in the spatial distribution of soil moisture or its spatial and linear relationships with key geophysical variables, such as temperature and precipitation [3,9,26].

In this research, we test the performance of three methods to gap-fill satellite-derived soil moisture in ESA CCI product version 4.5. Although version 4.5 includes a gap-filling strategy (as described above), this version still contains gaps across many regions of the world [22]. Our research aims to offer alternative strategies to provide spatially complete soil moisture estimates to complement the methods applied in the ESA CCI product, version 4.5 [21].

We tested three approaches. The first one is based on ordinary kriging (OK) spatial interpolation [27–29] to take advantage of the spatial autocorrelation of satellite-derived soil moisture on gridded surfaces. The second one performs regression kriging, which combines the principles of kriging interpolation and linear regression with covariates [27,30] that are used to solve kriging weights [31]. In this work, RK relies on the relation between soil moisture (response variable) with precipitation and minimum air temperature (explanatory variables). Our last approach is based on the application of generalized linear models (GLMs) to explore the relationship between soil moisture and the same explanatory variables integrated in our RK analyses. We tested these three methods because: (a) OK has the advantage of requiring solely spatial soil moisture information; (b) GLM has the advantage of benefiting from the inclusion of geophysical covariates (i.e., independent explanatory variables); and (c) RK incorporates both linear relationships and geospatial distribution of explanatory variables.

We focused our study over a region in the Midwestern United States (with abundant satellite-data estimates and in situ measurements) between 2000 and 2012. We evaluated the outcome of our gap-filling approaches with ground-truth information using in situ measurements from the North American Soil Moisture Database (NASMD) [15]. Our results show that the overall correlations between OK or RK with field data (i.e., NASMD) were slightly higher than those using GLM.

These results provide support for alternative techniques to complement other approaches aimed to gap-fill satellite-derived geophysical datasets [23,24] and highlight the potential of using geostatistical techniques. Furthermore, methods based on the spatial distribution of soil moisture, such as OK, which does not require information from geophysical covariates, are useful when covariate information (e.g., precipitation and air temperature) is missing in different regions across the world.

Section 2 provides a description of the region of interest as well as the parameters to select our time frame. Data acquisition, preprocessing, selection of the geophysical covariates, application of proposed gap-filling approaches, and the validation strategy are also described in Section 2. Section 3 describes the performance of OK, RK, and GLM techniques, as well as the results of cross-validation for the three models. Validation using reference correlation between original satellite data and ground-truth soil moisture information is also described in Section 3 and is compared with model outputs. Section 3 additionally shows the capability of our methods to reproduce the spatial soil moisture patterns shown by the original ESA CCI product. Section 4 proceeds with the discussion of our findings and their implications in providing spatially complete soil moisture information derived from ESA CCI satellite estimates from version 4.5. Section 5 summarizes the remarks of our work and their implications in providing soil moisture information for specific applications.

## 2. Materials and Methods

### 2.1. Region of Interest

The selected region of interest was an area of 465,777 km$^2$ (Figure 1a) centered in the state of Oklahoma (180,986 km$^2$) and covering some areas of surrounding states within Midwestern USA: Texas (159,489 km$^2$), Colorado (11,210 km$^2$), Kansas (61,343 km$^2$), Missouri (10,844 km$^2$), New Mexico (18,550 km$^2$), and Arkansas (23,356 km$^2$). The region of interest shows a variety of environmental conditions, both natural and human-driven, that allowed us to test the spatial performance of our gap-filling frameworks. This diversity mitigates bias due to specific environmental conditions (e.g., homogenous land cover, uniform topographic features), which are not the attention of this present study. The region of interest for this study was selected in response to the availability of ground-truth data in that area, mainly over Oklahoma, where mesonet [15] provides a robust set of historical soil moisture records [32]. Additionally, soil moisture data availability in northern Texas and the remaining areas in the region of interest are consistently represented by the NASMD. We highlight that the NASMD integrates data from several monitoring networks including mesonet [15].

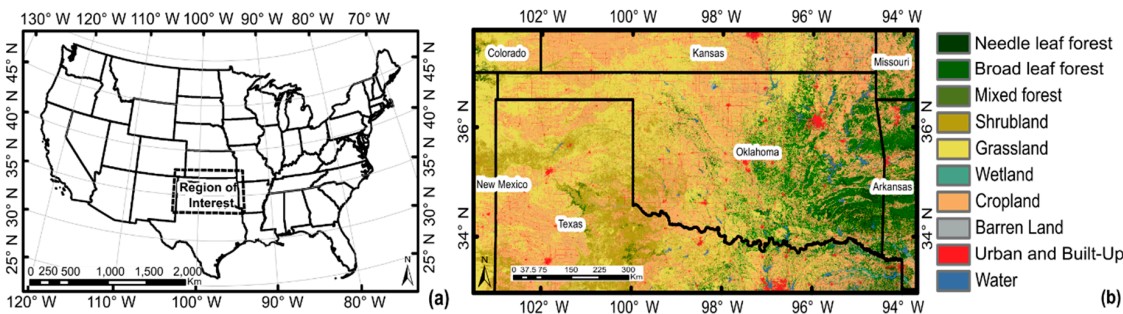

**Figure 1.** (**a**) Region of interest in the Midwestern USA, where soil moisture gap-filling methods were performed; (**b**) Land cover types over the region of interest (30 m), level 1 NALCMS classification [33].

The region of interest (Figure 1a) includes a wide variety of land cover types (Figure 1b) dominated by grassland (35.5%), cropland (31.9%), and shrubland (11.0%) in the central and western areas, whereas forested areas are mostly located in the eastern portion, distributed across needleleaf (2.2%), broadleaf (10.9%), and mixed forests (0.6%) [33].

*2.2. Data*

2.2.1. Satellite-Derived Soil Moisture

For this study, we used the ESA CCI soil moisture product version 4.5 (Table 1) that has gathered historical records from active and passive remote sensors [11,18,20]. This product provides soil moisture estimates at 0.25 degrees of spatial resolution on a daily basis, from November 1978 to December 2018 [20]. Active and passive sensors are combined by means of a weighted mean, being proportional to the signal-to-noise ratio (SNR) [20]. These ratios are estimated using triple collocation analysis, which is a method that estimates random error variances of three collocated datasets of soil moisture estimates [21]. In areas where no triple collocation analysis estimates are available, soil moisture values are estimated using a polynomial regression between the signal-to-noise ratios [20].

**Table 1.** Main characteristics of ESA CCI soil moisture version 4.5 [20].

| Title ESA CCI Soil Moisture Version 4.5 | |
|---|---|
| Release date | October 2019 |
| Available products | Active<br>Passive<br>Combined |
| Scatterometer sensors used | SMMR<br>SSM/I<br>TMI<br>AMI WS<br>ASCAT |
| Radiometer sensors used | Windsat<br>AMSR-E<br>AMSR2<br>SMOS |
| Available time span | November 1978 to December 2018 (Combined product)<br>August 1991 to December 2018 (Active product) |

The ESA CCI product was developed in collaboration with Vienna University of Technology (TU Wien) and focuses on the use of data derived from C-band scatterometers, such as European Remote Sensing Satellites (ERS-1/2) and METOP, as well as the use of data from multi-frequency radiometers such as the Scanning Multichannel Microwave Radiometer (SMMR), Special Sensor Microwave Imager (SSM/I), Microwave Imager (TMI), Advanced Microwave Scanning Radiometer (AMSR-E), and Windsat [3]. These sensors are characterized for the suitability for soil moisture retrieval [3].

Daily soil moisture global records from the ESA CCI product were acquired and then cropped to the region of interest. Daily estimates were merged into monthly soil moisture spatial layers using mean and median values; in this way, we tackled the lack of daily coverage in areas out of the satellites' swath. Monthly mean values initially reduced the number of gaps in daily products but still provided reliable information to identify spatial patterns and trends in our study period. These values then were used to explore their relationship with different geophysical covariates (Supplementary Material S1). Monthly values can describe soil moisture variability over a few weeks due to soil moisture memory effects, as water content derived from sudden excessive rainfall or lack of water onset can generate wetness or dryness conditions that might last for a couple of weeks [2].

An important step in preparing the soil moisture data for analysis is identifying the most relevant summary statistics, such as the mean or median. The median value is more useful when data are concentrated on a brief period of the month (because of long data gaps) with an uneven distribution of data [34]. However, mean monthly soil moisture values showed higher correlation with the tested set

of geophysical covariates (Supplementary Material S1). For our region of interest, Figure 2 shows the spatial distribution and number of soil moisture gaps (ESA CCI soil moisture version 4.5) during the study period (January 2000 to September 2012) where no mean values were calculated due to a lack of valid pixels. A pixel is considered valid when soil moisture estimates are available from the ESA CCI product over the region of interest. Figure 3 shows the number of gaps per monthly layer, regarding 741 pixels of 0.25 × 0.25 degrees in our region of interest.

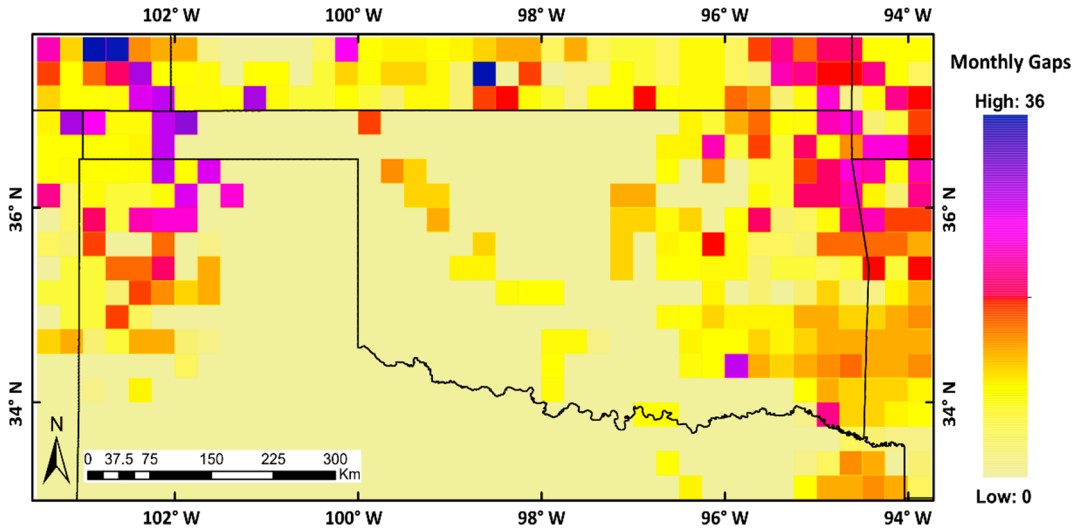

**Figure 2.** Number of gaps in monthly soil moisture estimates over the region of interest, derived from the ESA CCI product (version 4.5), January 2000–September 2012.

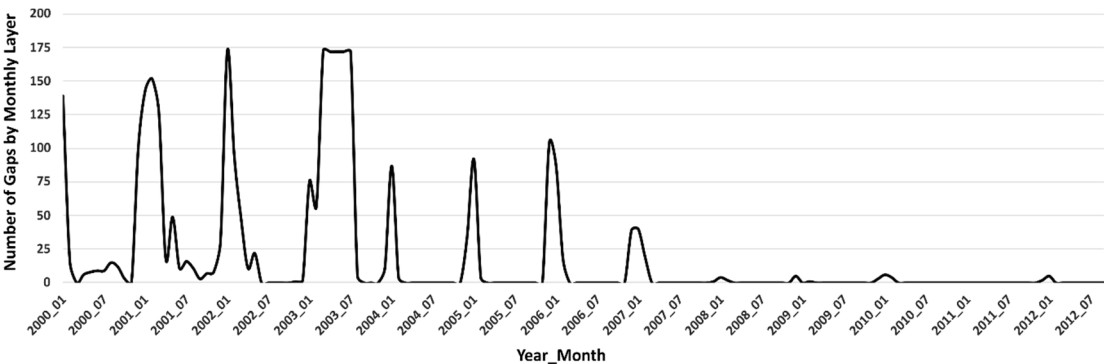

**Figure 3.** Distribution of gaps during the study period in monthly steps; each number represents the quantity of pixels without data, out of 741 pixels in the region of interest.

### 2.2.2. Soil Moisture Covariates

For RK and GLM gap-filling approaches, we explored the relationships between soil moisture and some geophysical variables. Monthly layers were generated for precipitation, atmospheric temperature, and static values of soil texture and the topographic wetness index (TWI). These selected variables are known to work as drivers for water input in soil [2,3].

Meteorological data were acquired at 1-km spatial resolution monthly layers produced by the Daily Surface Weather and Climatological Summaries (DAYMET) [35]. Total monthly precipitation and monthly averages of minimum and maximum air temperature raster layers from January 2000 to September 2012 were cropped to the region of interest, projected to the WGS84 Lat.–Long. coordinate system, and resampled to 0.25 degrees by means of the nearest neighbor method (ngb) [36].

Soil texture was obtained from the US soil survey geographic database [37], and we classified all classes into four general categories based on the texture triangle from the US Department of Agriculture

(USDA) [38]: coarse, medium, medium fine, and fine. Soil texture then was resampled to 0.25 degrees resolution using ngb [36]. We calculated TWI using SAGA GIS [39] with a digital elevation model at 250 meters resolution [27] and then resampled the output to 0.25 degrees using ngb [36]. Detailed information on the definition of geophysical variables for this work and their further processing are given in the Supplementary Material S1.

### 2.2.3. Validation Data

In order to establish a reference value that describes the spatial distribution pattern of soil moisture over our region of interest, we acquired records from the North American Soil Moisture Database (NASMD). NASMD provides the densest possible soil moisture network that integrates field measurements across North America [15]. By 2015, the NASMD had integrated 33 observation networks and two short-term soil moisture campaigns, providing ground-truth data for over 1800 observation sites in the USA, Canada, and Mexico [15]. Some of the densest regional networks integrated by NASMD offer soil moisture data in our region of interest (e.g., MESONET), and records at 5-cm depth, where the soil layer closely interacts with the atmosphere and it is sensed by satellites [40]. We extracted all information available from the NASMD over our region of interest that comprised records at 5-cm depth, from January 2000 to September 2012. Finally, we transformed these data to georeferenced point layers to be integrated in our ground-truth validation approach.

### 2.3. Gap-Filling Methods

Our first two gap-filling approaches were based on kriging interpolation (OK and RK). These techniques lead to high uncertainty over areas with very large continuous spatial gaps because they rely on the spatial autocorrelation of available data. Consequently, we also tested a third approach based on GLM to test the relationship between soil moisture and geophysical covariates. We clarify that the GLM approach does not depend on the spatial autocorrelation of available data.

The OK interpolation strategy depends solely on the separation distance between sampled locations and not on an absolute position [29]. This offers a feasible strategy to fill spatial gaps in areas where no other information is available to be included in similar interpolation methods such as cokriging or regression kriging. This is the most popular among all kriging methods, as it works in almost any situation and its assumptions are easily filled [29].

Regression kriging also depends on the spatial location of soil moisture values but incorporates the location of information from covariates as well [27]. Regression kriging yields to a better representation of the spatial patterns depicted by the covariates known as be correlated with the response variable [30].

Generalized linear models (GLMs), as an alternative approach, represent multivariate regression models [41]. In this approach, we assume linear relationships between the dependent variable (soil moisture) and the predefined covariates (precipitation, minimum air temperature) before considering relationships that are more complex. These relationships have also been explored in previous studies of soil moisture derived from field measurements, integrating predictors such as vegetation indices, precipitation, and temperature [42,43]. However, GLM represents an approach that can be applied to satellite-derived soil moisture estimates to fill spatial gaps over large areas.

Soil moisture spatial-gaps in the region of interest are not always sufficient to test interpolation methods, as in some months there are no gaps over the region of interest. Thus, we decided to randomly remove valid data from each soil moisture monthly layer as well as their correspondent locations on the geophysical covariates layers. Therefore, OK, RK, and GLM were performed on 100%, 75%, and 50% of available valid pixels in each month, similar to gap-filling analyses in previous studies [23].

The overall process for soil moisture prediction (Figure 4), derived from the proposed modeling techniques, was evaluated using cross-validation and ground-truth data from the NASMD available from January 2000 to September 2012. An extensive description of the workflow and a sample process for one month are provided in the Supplementary Material S2.

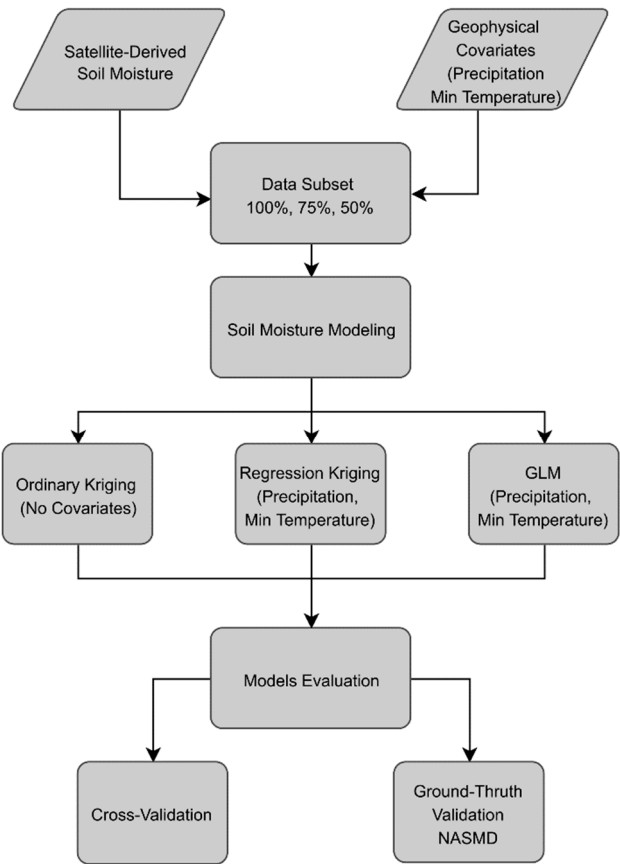

**Figure 4.** Workflow for soil moisture modeling and the gap-filling over the region of interest, regarding 100%, 75%, and 50% of available valid pixels in each monthly layer. Cross-validation as well as ground-truth validation is also described.

### 2.3.1. Ordinary Kriging

OK was performed using the AutoMap package developed for the R statistical platform [44]. By means of the autofit-variogram tool, the best-fitted variogram model was automatically selected to generate independent predictions over each month. Five different variogram models (i.e., spherical, exponential, Gaussian, Matern and Stein's parameterization) were evaluated, and the one with the smallest residual sum of the squares was selected [44]. The prediction of values at unsampled locations is the linear combination of N variables, as expressed in Equation (1):

$$Z(u) = \sum_{i=1}^{N} \lambda_i \, Z(u_i) \tag{1}$$

where $\lambda_i$ represents the original weighted values. Weights are calculated as a function of the distance between sampled and unsampled locations to be predicted. The weight sum must be equal to 1, thus estimations fulfill the unbiasedness requirement [45].

Derived from OK spatial interpolation, predicted values as well as their standard errors were obtained for each month, derived in three different cases from 100%, 75%, and 50% of available valid pixels. We applied 10-fold cross-validation [44] to OK outputs for the above-mentioned percentages of valid pixels using autoKrige.cv [44]. Finally, we assessed the spatial dependence found in each monthly layer using the nugget–sill ratio. Ratios of at most 0.25 represented strong spatial dependence; between 0.25 and 0.75, moderate spatial dependence; and at least 0.75, weak spatial dependence, as previously reported [46].

### 2.3.2. Regression Kriging

RK was performed with the R package GSIF [47], using the function fit.regModel. Individual regression models were fitted to each monthly layer, incorporating monthly precipitation and minimum temperature data from DAYMET [35]. We combined regression on soil moisture data and the preselected geophysical covariates with simple kriging of the regression residuals [31]. GSIF tools allowed us to select different regression techniques (e.g., random forest, GLM, quantile regression forest). We selected GLM to make RK a hybrid approach between our two other proposed methods (i.e., OK and GLM). In RK, a spatial trend is assumed instead of stationarity across the region of interest. Based on the residuals of the identified trend in regression analysis, spatial interpolation is applied through OK. Prediction over unsampled locations is equal to the estimated trend plus the error prediction as expressed in Equation (2):

$$Z(x) = m(x) + \varepsilon(x) \tag{2}$$

where $Z(x)$ is the target variable to be predicted, $m(x)$ is the trend (explanatory power) identified from the relationship with geophysical covariates, and $\varepsilon(x)$ represents the stochastic residuals. Unlike OK, in RK, the trend is no longer constant, but is a function of the explanatory variables [48].

As we did for OK, we derived predicted values and associated error based on 153 months, using 100% of available data, as well as 75% and 50%; this yielded 459 predicted soil moisture layers. Then, 10-fold cross-validation was performed, and nugget–sill ratios were calculated as in the OK approach to identify the level of spatial dependence [46] depicted in each monthly layer.

### 2.3.3. Generalized Linear Models

For GLM, we first tested the overall correlation between soil moisture (monthly mean and median values) and each one of the geophysical covariates (monthly precipitation, monthly maximum and minimum air temperature, soil texture, and TWI). Secondly, we extracted a time series for each valid pixel along the 153 monthly soil moisture layers and tested the pixel-individual correlation with each one of the covariates. Finally, we calculated the correlation coefficients of all valid pixels available for each monthly layer with the corresponding temporal layer for each one of the covariates. Based on these analyses, we established that the spatial values of mean monthly precipitation and minimum air temperature were the variables with the highest absolute correlation coefficient with mean monthly soil moisture (Supplementary Material S1). These geophysical covariates were used to predict soil moisture based on GLM, as shown in Equation (3):

$$Y_i = \beta_0 + \beta_1 X_{i1} + \beta_2 X_{i2} + \varepsilon_i \tag{3}$$

where $Y_i$ represents the response variable, $X_{i1}$ and $X_{i2}$ represent the predictor variables, $\beta_0$, $\beta_1$ and $\beta_2$ are the parameters of the model, and $\varepsilon_i$ is the error term [41].

Predictions were also performed for the three predefined subsets (100%, 75%, and 50%) of available valid data over the region of interest in each month of the study period. We used the GLM tool from the caret statistical package in R [49] to generate independent models for each month, as well as a 10-fold cross-validation process. For this purpose, we used 75% of the data in each independent monthly dataset as training data and 25% as test data.

### 2.4. Ground-Truth Validation

### 2.4.1. Reference Correlation between NASMD and Satellite-Derived Soil Moisture

First, we established a reference correlation value between original satellite-derived soil moisture and data from the NASMD. We extracted all available data from NASMD over the region of interest for each month during the study period and calculated the mean monthly value of soil moisture at 5-cm depth for each field station, thus capturing as much variation as possible from the upper soil layers sensed by the satellites. We tested the correlation between satellite-derived values over each

spatially correspondent pixel with soil moisture information derived from the NASMD. This process was performed over the layers using 100%, 75%, and 50% of available valid pixels. When there was more than one NASMD station within one corresponding pixel of satellite-derived soil moisture, every station value from within the pixel area was accounted for in the correlation analysis with the satellite data. Overall, we used data from 157 stations in the months with the highest availability of field soil moisture records. The use of all NASMD available stations allowed us to retain the overall observation-estimation pairs. Figure 5 shows the distribution of available NASMD stations over the region of interest for the entire study period. Figure 6 shows the number of NASMD stations used in each month to validate the outputs of our models. Across the entire study period, all available stations provided 19,007 points to compare satellite-derived soil moisture estimates and ground-truth data.

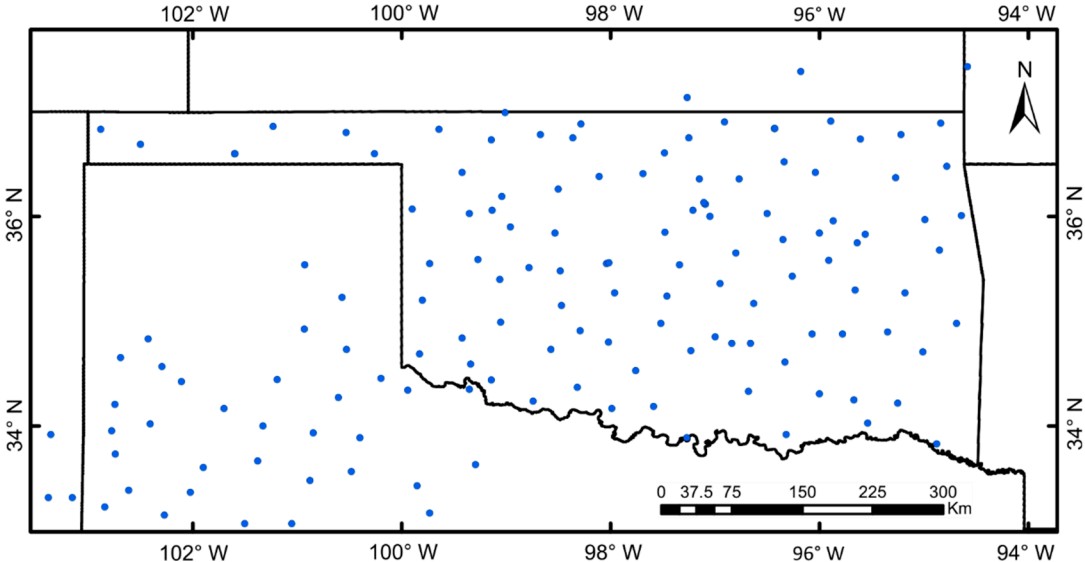

**Figure 5.** All NASMD stations available for the study period (157). Stations are broadly distributed over Oklahoma and northern Texas; however, ground-truth data are scarce over the surrounding states within the region of interest.

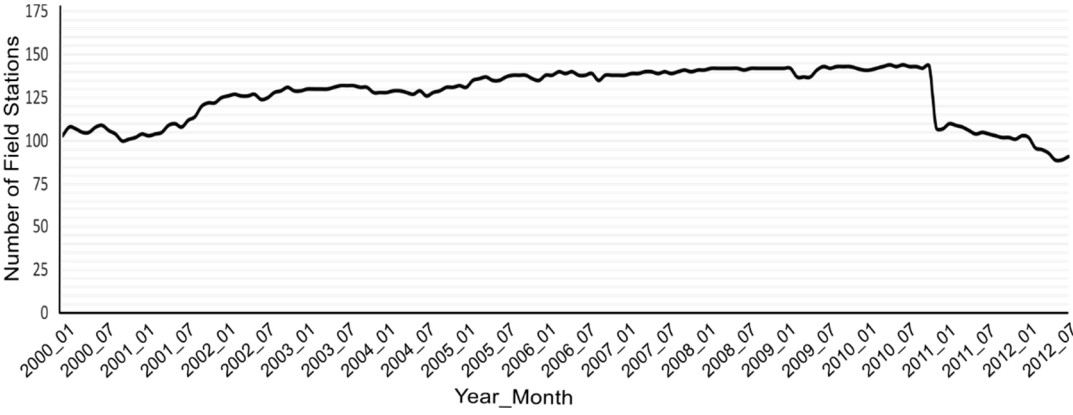

**Figure 6.** Number of ground-truth stations available per month for validation.

### 2.4.2. Correlation between Predicted Soil Moisture and NASMD

In order to validate our soil moisture predicted values, we looked for the closest similar correlation coefficient from our outputs and the NASMD to the correlation coefficient between the original ESA CCI estimates with NASMD, thus repeating the same value of a satellite estimate or predicted value for each field station that is located within the same cell. In this way, we take advantage of as much

validation information as possible over our region of interest. We followed the same approach as in Section 2.4.1 to evaluate the soil moisture values derived from the modeling approaches with the NASMD. This allowed us to evaluate 19,411 pixels where we calculated the overall correlation coefficient (all months) and monthly correlation coefficients.

## 3. Results

### 3.1. OK and RK Models Selected for Soil Moisture Predictions

Variograms using Stein's parameterization [50] were the most common in OK across the 459 monthly layers (n = 402). Exponential (n = 53), spherical (n = 3), and Gaussian (n = 1) were used in a substantially lower number of predicted soil moisture layers. RK was based on exponential variogram models in all cases (459 monthly layers), regardless the percentage of valid data used (100%, 75%, or 50%). We found strong spatial dependence in 416 of the monthly layers (nugget–sill < 0.25) and moderate spatial dependence in the remaining 43 layers (0.25 < nugget–sill < 0.75) when using OK (Figure 7a). On the other hand, we found strong spatial dependence in 253 monthly layers out of 459 and moderate spatial dependence in 206 when using RK. The RMSE for predicted soil moisture layers with OK showed that Stein's parameterization [50] and spherical models had smaller minimum values. However, we found that the RMSE values were more distributed in Stein's parameterization than in spherical models. RK with exponential models had a higher RMSE value than OK, but the error distribution was less spread, with just a few extreme values (Figure 7b).

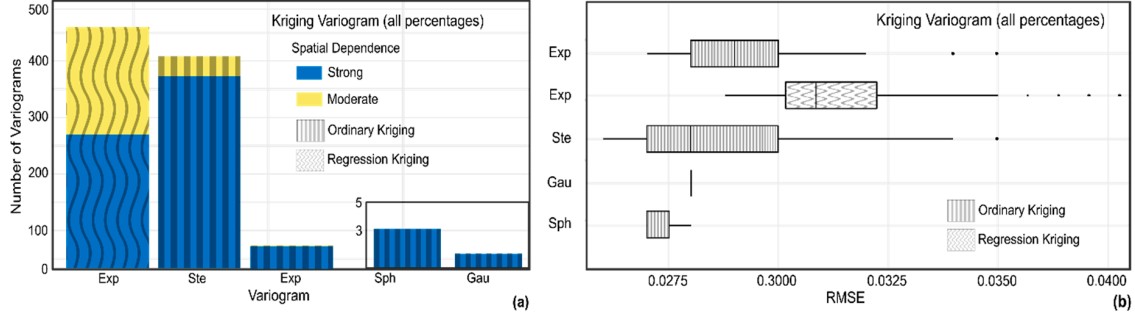

**Figure 7.** (**a**) Most fitted variogram models used to predict soil moisture; 459 models generated for both OK and RK in 153 monthly layers derived from all percentages (100%, 75%, and 50%) of valid pixels; (**b**) Boxplots of the RMSE for each predicted layer using the selected variograms.

### 3.2. Cross-Validation of Predicted Values

Overall, the three models had good cross-validation results, but OK and RK had consistently higher correlation coefficients and lower RMSE (Table 2). However, OK had slightly better performance than RK when a different percentage of available data was used.

**Table 2.** Cross-validation outputs for OK, RK, and GLM, and all predicted and observed values along the 153 monthly layers.

| Method | Percentage of Data | Correlation | RMSE |
|--------|--------------------|-------------|------|
| | 100% | 0.886 | 0.029 $m^3m^{-3}$ |
| **OK** | 75% | 0.886 | 0.029 $m^3m^{-3}$ |
| | 50% | 0.886 | 0.029 $m^3m^{-3}$ |
| | 100% | 0.886 | 0.029 $m^3m^{-3}$ |
| **RK** | 75% | 0.878 | 0.030 $m^3m^{-3}$ |
| | 50% | 0.869 | 0.031 $m^3m^{-3}$ |
| | 100% | 0.711 | 0.044 $m^3m^{-3}$ |
| **GLM** | 75% | 0.709 | 0.044 $m^3m^{-3}$ |
| | 50% | 0.709 | 0.044 $m^3m^{-3}$ |

Additional cross-validation between predicted and observed values by month (January to December) was reported using Taylor diagrams (Figure 8), which simultaneously report the correlation coefficient, normalized standard deviation, and centered root mean squared error [51]. The Taylor diagrams [52] consistently showed that OK and RK had a higher correlation coefficient and lower centered RMSE and standard deviations, and consequently, were closer to the observations. These results were consistent regardless of the percentage of available data used. Overall, OK had a consistent correlation coefficient of 0.886, whereas RK ranged from 0.869 to 0.886 as the percentage of data to model values was lower. Finally, GLM values ranged between 0.711 to 0.709 with a lower percentage of valid data. Centered RMSE values between observed and predicted values with OK were consistently 0.029, RK ranged between 0.029 and 0.031, and GLM values were 0.044 $m^3m^{-3}$ in all cases.

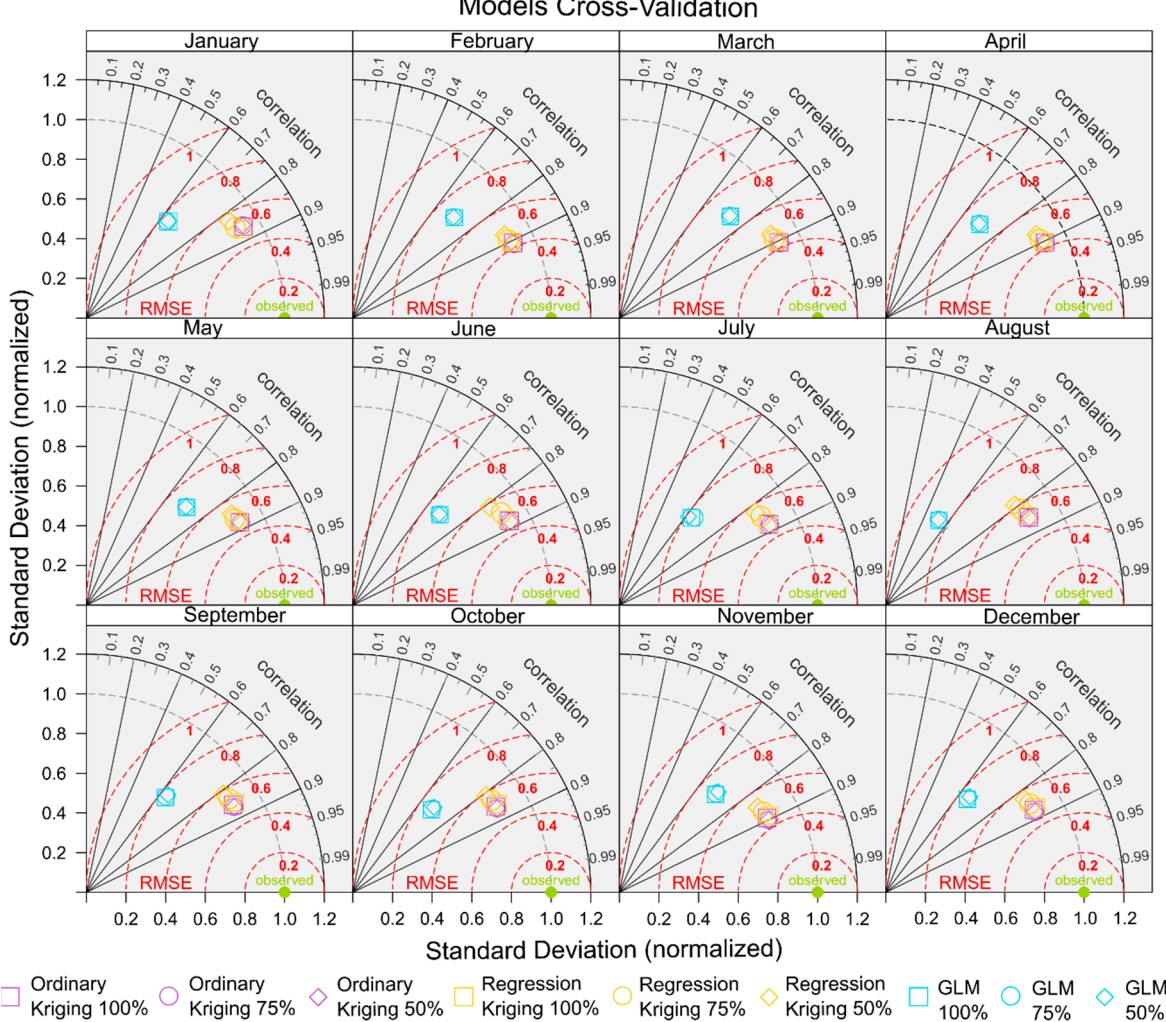

**Figure 8.** Taylor diagrams based on cross-validation by month across years; OK, RK, and GLM using 100%, 75%, and 50% of available valid data. Standard deviation and RMSE values for each output were normalized using the standard deviation of the observed values.

### 3.3. Ground-Truth Validation with NASMD

We found an overall correlation coefficient of r = 0.523 and an RMSE of 0.093 $m^3m^{-3}$ between the original ESA CCI data and the available NASMD stations across the study period (153 months). These values served as a baseline and showed that values generated using OK with 100% and 75% of valid data were closer to the reference than those using RK and GLM (Table 3).

**Table 3.** Overall correlation coefficients between all ground-truth validation points and the CCI soil moisture product, as well as gap-filled outputs. Percentages show the data subset used to predict soil moisture values over the region of interest.

| Method | Percentage of Data | Correlation | RMSE |
|:---:|:---:|:---:|:---:|
| **CCI** | 100% | 0.558 | 0.069 $m^3m^{-3}$ |
| **OK** | 100% | 0.579 | 0.067 $m^3m^{-3}$ |
| | 75% | 0.575 | 0.067 $m^3m^{-3}$ |
| | 50% | 0.569 | 0.067 $m^3m^{-3}$ |
| **RK** | 100% | 0.582 | 0.067 $m^3m^{-3}$ |
| | 75% | 0.582 | 0.067 $m^3m^{-3}$ |
| | 50% | 0.571 | 0.067 $m^3m^{-3}$ |
| **GLM** | 100% | 0.475 | 0.070 $m^3m^{-3}$ |
| | 75% | 0.475 | 0.070 $m^3m^{-3}$ |
| | 50% | 0.475 | 0.070 $m^3m^{-3}$ |

We explored the temporal dynamics of the correlation coefficients and RMSEs by month throughout the study period. Figure 9 shows the R-squared values between the monthly correlation coefficients from ground-truth data and CCI products and the coefficients from ground-truth data and predicted values by our proposed methods (OK, RK, GLM). RMSE is reported in the same manner (Figure 9). OK correlation coefficients using 100% of available valid data with ground-truth data are the closest to the correlation coefficients used as a reference between validation data and the CCI product (Figure 9a). However, RK correlation coefficients show higher consistency when compared with the reference correlation coefficients across different percentages of available valid data (Figure 9b). In contrast, GLM outputs show lower general R-squared values between the outputs and the reference and are loosely fitted to the regression line (Figure 9c). In a similar way, R-squared values between RMSE from the CCI product and ground-truth data, as well RMSE from model outputs and ground-truth data, show closer relation for OK (Figure 9a) and RK (Figure 9b) outputs rather than for GLM (Figure 9c). Nevertheless, OK shows slightly better results than RK.

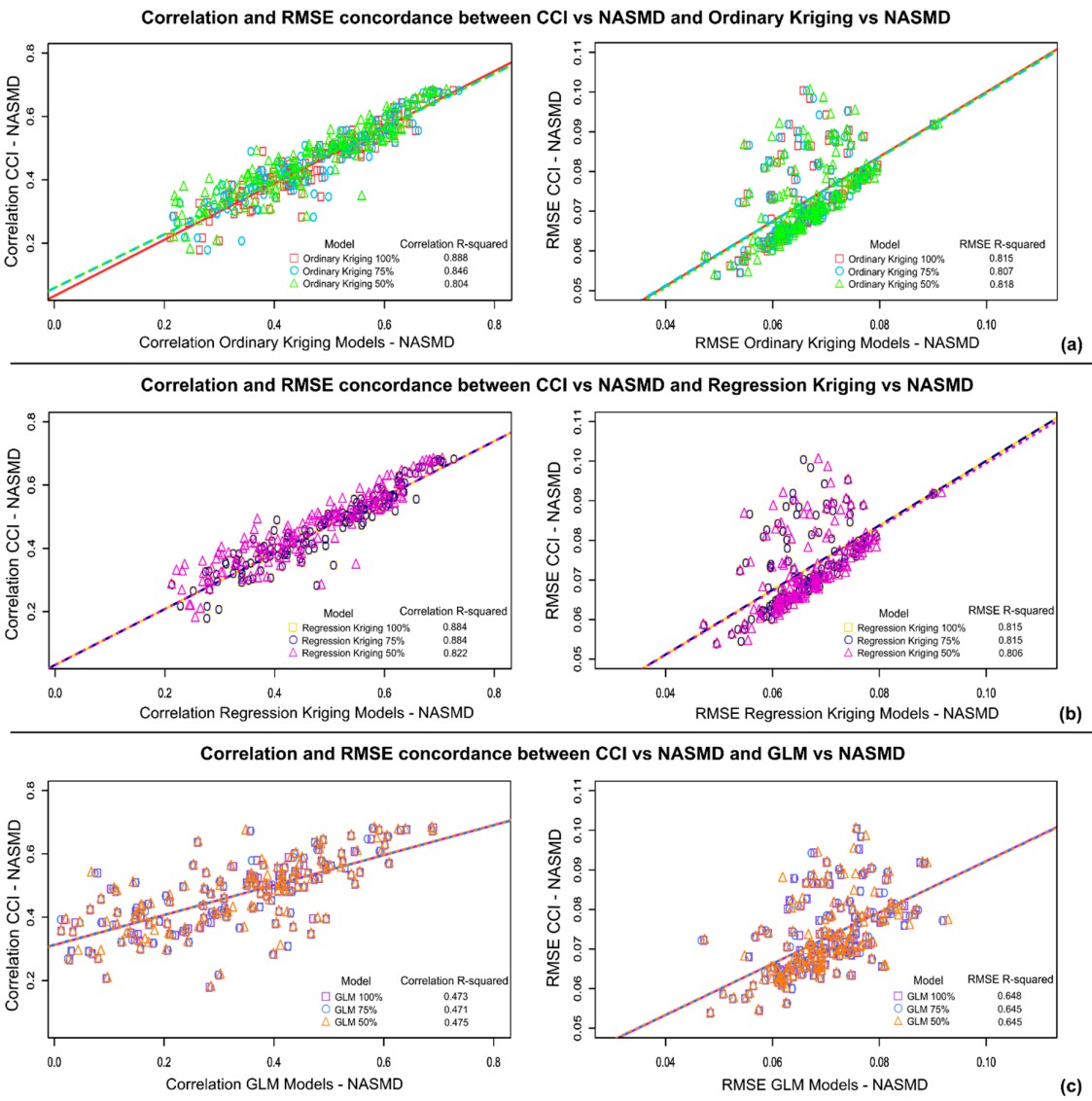

**Figure 9.** R-squared values showing the concordance between the reference validation (CCI-NASMD) and the validation of proposed gap-filling methods with NASMD. Each point represents one month, using 100%, 70%, or 50% of the available data, with the percentage indicated by the shape used. (**a**) Validation of ordinary kriging with NASMD, with correlation on the left and RMSE on the right; (**b**) Validation of regression kriging with NASMD, with correlation on the left and RMSE on the right; (**c**) Validation of GLM with NASMD, with correlation on the left and RMSE on the right. Regression lines between correlated datasets are shown in each plot.

*3.4. Spatial Gap-Filling Performance of Modeling Methods*

The comparison between the outputs of our modeling methods in contrast with the original ESA CCI soil moisture product shows that OK and RK approaches better reproduce the spatial pattern captured by satellite estimates. Figure 10a shows the mean soil moisture estimates from the ESA CCI product version 4.5 derived from 153 monthly layers in our region of interest, without any gap-filling technique. In comparison to the original spatial distribution of soil moisture, OK visually shows more similar patterns, independent of the percentage of valid pixels used for modeling (Figure 10b–d). RK visually shows very similar spatial patterns (Figure 10e–g) as OK. However, both methods, OK and RK, are challenged by extreme low and high values included in the original satellite product.

Conversely, GLM shows a lower performance in reproducing soil moisture spatial patterns, regardless of the percentage of valid pixels included in the modeling process (Figure 10h–j).

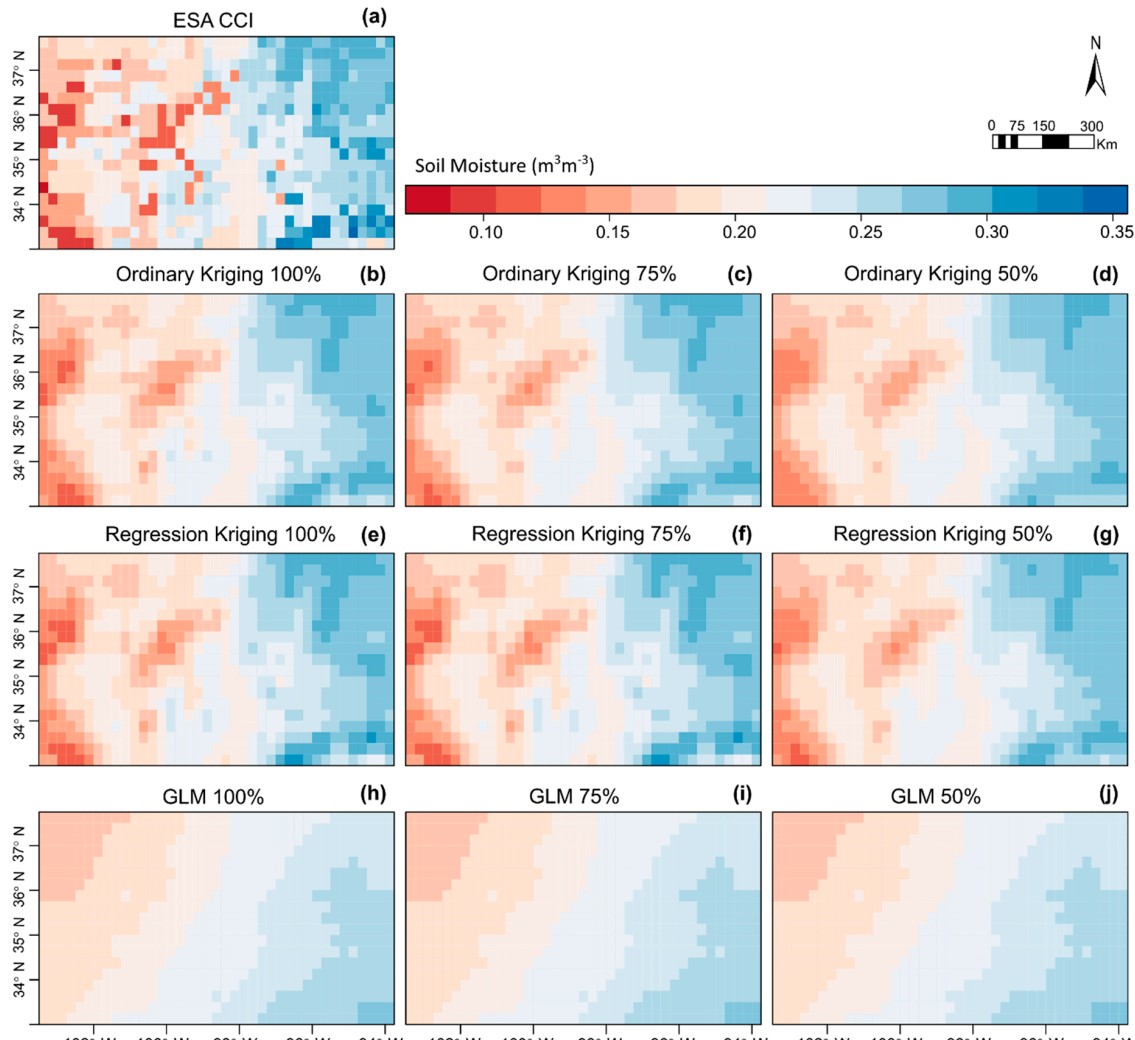

**Figure 10.** Mean soil moisture values during the study period (January 2000–September 2012) over the region of interest. (**a**) Mean values of original ESA CCI soil moisture estimates; no gap-filling methods applied; (**b**) Soil moisture mean values modeled using OK and 100% of available valid data; (**c**) Soil moisture mean values modeled using OK and 75% of available valid data; (**d**) Soil moisture mean values modeled using OK and 50% of available valid data; (**e**) Soil moisture mean values modeled using RK and 100% of available valid data; (**f**) Soil moisture mean values modeled using RK and 75% of available valid data; (**g**) Soil moisture mean values modeled using RK and 50% of available valid data; (**h**) Soil moisture mean values modeled using GLM and 100% of available valid data; (**i**) Soil moisture mean values modeled using GLM and 75% of available valid data; (**j**) Soil moisture mean values modeled using GLM and 50% of available valid data.

Finally, we found that the density distribution describing the mean soil moisture values during the study period in the original ESA CCI was better reproduced by the OK and RK approaches. The performances of OK and RK were similar, either using 100%, 75%, or 50% of available valid data (Figure 11a,b). In contrast, the GLM density distribution substantially deviated from the values of the original ESA CCI product (Figure 11c).

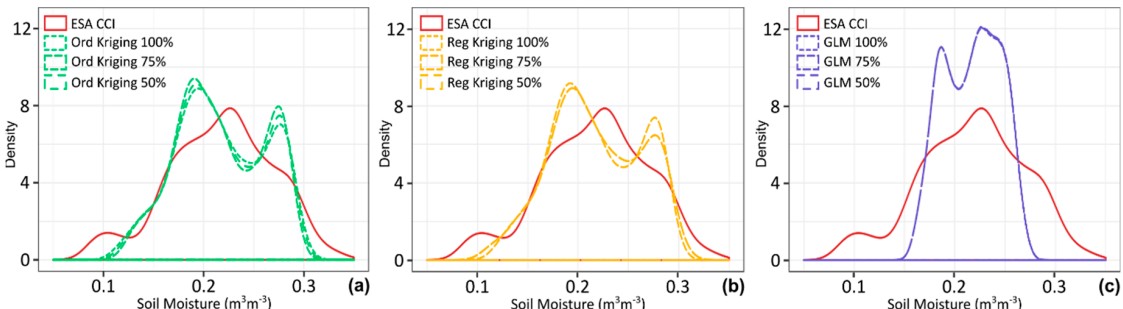

**Figure 11.** Density distribution of mean soil moisture values during the study period for 741 pixels over the region interest. (**a**) ESA CCI and modeled data using OK with 100%, 75%, and 50% of available valid data; (**b**) ESA CCI and modeled data using RK with 100%, 75%, and 50% of available valid data; (**c**) ESA CCI and modeled data using GLM with 100%, 75%, and 50% of available valid data.

## 4. Discussion

Our results showed that the OK, RK, and GLM techniques could be used as alternative approaches to gap-filling in soil moisture data derived from the ESA CCI product version 4.5. Our proposed methods can be used either in conjunction with geophysical covariates such as precipitation and temperature or using solely the spatial distribution of soil moisture estimates derived from the ESA CCI product. Furthermore, our results show that spatial patterns and temporal relations between satellite and ground-truth data are better preserved by using OK and RK, but we show the applicability of the GLM approach. The benefit of using different approaches would depend on the spatial structure of the missing data and the availability of covariates for applying OK, RK, or GLM approaches.

Precipitation and minimum air temperature were the strongest correlated environmental covariates with soil moisture (Supplementary Material S1). These relationships are likely influenced by the grid size (0.25 degrees), as the spatial influence of precipitation and air temperature represents regional and mesoscale climatic patterns [53]. Previous research showed that increasing spatial resolution yields more detail in the meteorological information but limited impacts on its forecasting skill [54]. It is known that from the plot to watershed scale, soil texture and topography are highly correlated with soil moisture [2,3], but these relationships may change at the coarse scale of the ESA CCI soil moisture product. Thus, these features were not included as geophysical covariates in our GLM or RK approaches.

Overall, our results provide support for OK, RK, and GLM as techniques to gap-fill spatial missing values of satellite-derived soil moisture products. However, overall performance indicates that OK and RK represent more reliable methods for soil moisture gap-filling in comparison with GLM. Previous studies have compared the advantages of OK and RK for interpolation of spatial soil moisture and other soil properties [27,55–58] but most analyses have been performed for spatial interpolation of soil properties based on field data [26,58–60]. OK has been regarded as an unbiased linear estimator [45], and our results support it as a feasible approach due to the spatial scale of the original ESA CCI estimates (0.25 degrees) under the gap scenarios tested in this work. At this coarse scale, soil moisture values represent a quasi-continuous matrix that meets basic assumptions of kriging analysis such as stationarity [45] and spatial dependence [58]. OK also incorporates spatial autocorrelation by using the variogram and providing the error variance estimation from predicted values, offering some advantages over deterministic methods such as inverse distance weighting (IDW), which may create noisy fields in interpolation processes. Similar to other kriging methods, OK is an exact interpolator, which ensures that values at sampled locations are exactly preserved. Thus, we aim to fill the spatial gaps by modeling the entire region of interest, while preserving original values where data existed previously. Additionally, OK performs value predictions based solely on spatial data distribution, offering a suitable approach in cases where no well represented covariates datasets are available over the region of interest, and it compensates for data clustering [61]. Additional evidence in support

for OK is the fact that the nugget–sill ratio was less than 0.25 in 99% of the fitted variograms, which implies strong spatial dependence as discussed elsewhere [46].

RK on the other hand has been widely used to incorporate covariates to build a regression model with soil properties [27,62–64]. Whereas some authors do not find a better performance of OK in comparison with RK for the prediction of soil properties [27,62,63], our results support the use of RK, as it performed similarly to OK in our region of interest. As a hybrid method, RK has the advantage of incorporating spatially explicit information known to be correlated to the response variable [27,65]. The explicit correlation between soil properties and geophysical covariates provided good results when using terrain parameters [62,66,67] or other variables such as bare soil from remotely sensed sources, crop yield, temperature, and precipitation data [64,68] as predictors. Other authors highlight that RK performance depends on the relationships between soil and environmental factors [63,65]. This could explain the similar performance between OK and RK in our region of interest, as our selected covariates seem to account for similar influence at the coarse spatial scale of the ESA CCI product. Based on the spatial dependence depicted by the nugget–sill ratio in the fitted variograms using RK, we postulate that regardless of the similar performance using OK, our selected covariates did not have a consistent strong spatial dependence. Based on nugget–sill ratios, RK showed strong spatial dependence in 55% of the fitted variograms, while 45% showed moderate dependence when using the thresholds previously discussed [46]. Finally, it is possible that RK may not accurately describe the spatial patterns of soil properties when using coarse resolution geophysical covariates, but these covariates might help to improve prediction accuracy [30]. Thus, the incorporation of covariates may depend on the actual spatial dependence observed when modeling variograms using both OK and RK techniques.

The GLM approach allowed us to explore the most evident relationships between soil moisture in the upper layer of soil and the geophysical covariates that we found to be better correlated (Supplementary Material S1). We followed a parsimonious principle by means of the GLM technique, applying the simplest model with the fewest assumptions before assuming relationships that are more complex. This parsimonious reasoning and its applications to multivariate models have been explored in other studies [69].

The evaluation of our three approaches (OK, RK, and GLM) by means of cross-validation regarding their prediction capacity for actual satellite data shows similar correlation coefficients as those reported by [59] in the spatial interpolation of soil moisture and similar RMSE as reported by [58] for other soil properties. The cross-validation technique has been commonly used in other similar studies [58,59] and offers initial insights into modeling techniques without considering ground-truth data for validation. Our cross-validation strategy showed that OK and RK better predicted soil moisture values compared with GLM, in spite of pixel removal at different percentages. Regarding cross-validation for monthly grouped values, OK, RK, and GLM did not show an evident bias due to seasonality, as monthly correlation coefficients and RMSE values systematically describe the same patterns found when using data from the entire study period in a single dataset.

In spite of cross-validation results, ground-truth validation was performed to evaluate the suitability of each method (OK, RK, and GLM) to predict missing values in the ESA CCI product. We acknowledge the conceptual challenge of this data matching and the need of balancing ground-truth information in order to be representative of satellite-derived estimates. Representativeness challenges in validation of the ESA CCI product have been also acknowledged previously [40]. Two main problems are identified [40]: (1) Satellite sensors retrieve ground information from the upper soil layer (0.5–5-cm depth); this layer is directly exposed to the atmosphere; therefore, its physical characteristics may differ from the information provided by soil moisture sensors placed at 5-cm depth or deeper. Thus, satellite estimates represent a more variable soil layer, different from soil at deeper layers. (2) Even a spatially extensive soil moisture network cannot cover any area widely enough to provide scaling representativeness between point-scale measurements and satellite estimates. Field measurements depict soil characteristics in the range of a few square decimeters, while satellite products commonly

cover a few kilometers per pixel (~27 km pixel sizes in the ESA CCI product). Additionally, other authors suggest the soil moisture representativeness, on a grid-scale domain, may be described based on three different methods [30]: (1) empiric methods, averaging all points within each single grid-cell; (2) upscaling methods based on time information; and (3) spatial interpolation by means of kriging methods to assign individual values to each center point in the grid-cell domain. In this regard, our work does not aim to provide strategies of accuracy assessment between field measurements and satellite estimates as explored by [70]. We seek to reproduce the spatial soil moisture patterns expressed by the satellite-derived soil moisture and its actual correlation with ground-truth data with the ultimate goal to gap-fill missing information.

As proposed by [57], the selection of reliable ground-truth stations and the definition of core validation sites (CSV) represent a step forward in the evaluation of remotely sensed soil moisture. However, regarding the limited availability of ground stations providing soil moisture information, we integrated all available ground-truth data for our region of interest instead of defining CSV. In this way, we took advantage of all available field soil moisture records over the region of interest. This approach might introduce uncertainty, as neighboring stations within the same 0.25 degrees pixels in some cases could be affected by different moisture conditions in large areas. However, as our approach aims to reproduce the spatial distribution of soil moisture showed by the satellite estimates based on the correlation with ground-truth data, we aim to retain all the variation offered by NASMD stations.

In order to define the best-tested soil moisture prediction model to fill the gaps in the ESA CCI product, version 4.5, correlation found with ground-truth data was set as a reference for our proposed models in every month of the study period. This yielded a more specific way to validate our proposed methods regarding different soil moisture estimate conditions in every month of the ESA CCI product. Given that our research aims to complete spatial information of ESA CCI, reference correlation coefficients helped us to define which model best reproduces the spatial pattern of the original product. OK and RK showed better results than GLM, as we found the higher the number of valid pixels to shape the variogram parameters, the closer the correlation coefficient to the reference. Furthermore, OK and RK performance does not significantly decrease even though valid pixels are artificially removed. On the other hand, GLM correlation with ground-truth data showed less similar values to the reference, independent of the percentage of valid pixels removed.

Given that the OK, RK, and GLM performance for our region of interest is not that different, GLM can be an alternative approach in similar regions where satellite-derived soil moisture estimates are spatially scarce or highly clustered, as GLM relies more on predictor availability than on spatial distribution. Besides, when OK and RK do not meet the best requirements, GLM can use input data from robust meteorological datasets [71,72] to obtain the geophysical covariates that we used in our analysis. Based on the correlation coefficient between the ESA CCI soil moisture product and NASMD ground-truth data, we found that OK and RK consistently better reproduce reference correlation coefficients and RMSE values. Nevertheless, GLM correlation coefficients and RMSE values with NASMD do not significantly decrease from the reference, which still makes this method an alternative approach to gap-filling. Finally, the analysis of the mean soil moisture spatial patterns during the study period showed that OK and RK outputs consistently better reproduced the spatial patterns in the original ESA CCI product. This can be visually distinguished on the mean soil moisture maps, as well as in the density distribution of the original product in comparison with OK, RK, and GLM outputs.

We acknowledge that OK and RK represent the best-tested methods for soil moisture prediction and gap-filling of the ESA CCI product over our region of interest, based on the analysis of the monthly mean values from January 2000 to September 2012. The application of these methods in other regions and under different conditions should consider availability and distribution of soil moisture estimates since in large discontinuous areas, stationary can be wrongly assumed, yielding high uncertainty in predicted values. We recognize the need to explore RK models at finer spatial scales, where linear relationships with geophysical covariates such as those explored in the Supplementary Material S1 might be stronger. In future research, it is necessary to explore ESA CCI gap-filling over larger areas

such as the conterminous United States, where well spatially represented meteorological datasets are available and different scenarios of gaps distribution can be tested. Daily data must be also incorporated, as this is the temporal resolution in which original soil moisture estimates are delivered, thus opening the possibility to operationally fill the gaps in the original soil moisture estimates provided by the SA CCI soil moisture product (version 4.5). These implementations represent an upscaling need in computational capacities; therefore, high-performance computing (HPC) techniques must be considered.

## 5. Conclusions

For the region of interest, linear geostatistics techniques offer a suitable approach to fill the soil moisture spatial gaps of the ESA CCI product (version 4.5). Although the current version of the product follows different strategies to fill data gaps, our research highlights the incorporation of the spatial distribution of soil moisture, as well as the use of geophysical covariates to model missing values. Selected geophysical covariates to model soil moisture in this study, i.e., precipitation and minimum air temperature, can be easily integrated due to their historical availability across larger regions, e.g., the conterminous United States (CONUS). The selected region of interest provided a spatially extent set of valid pixels from January 2000 to September 2012, which allowed us to test our proposed methods under different scenarios of gap presence, due to natural conditions as well as artificial pixel removal.

The ordinary kriging method does not need to use any additional covariates, as it is built upon the spatial distribution of soil moisture data; on the other hand, RK benefits from relationships with geophysical covariates such as the ones explored in this work. However, these methods can be inconclusive over areas where reference data are highly sparse or clustered (i.e., data scenarios where we found weak spatial structure for satellite soil moisture). Generalized linear models, on the contrary, might offer an alternative to spatially model soil moisture and fill the gaps in the ESA CCI product, though their performance was lower than that of OK and RK in our region of interest. Soil moisture at a coarse scale can be significantly correlated with covariates such as precipitation and minimum air temperature, which can be easily inputted by predicting models over most of CONUS and other regions around the world.

Derived from cross-validation for each method and specific percentage of available data, the three proposed methods—ordinary kriging, regression kriging, and generalized linear models—showed a significant prediction performance with respect to soil moisture data. However, as we intended to reproduce the soil moisture spatial patterns of the ESA CCI product and its relationship with ground-truth soil moisture data, we considered field validation as the best approach to find the most suitable gap-filling method.

Besides offering information for a wide variety of applications by itself, spatially complete soil moisture information covering large areas can also be related to point-based soil moisture networks to jointly monitor ecological processes. Thus, gap-filled data can yield a better understanding of the role of soil moisture in water and carbon cycles, with important implications in plant and soil respiration, or plant growth, therefore influencing our capacity to predict climate change signals in soil moisture estimates from the regional to the global scale.

**Supplementary Materials:** The following are available online at http://www.mdpi.com/2072-4292/12/4/665/s1, Supplementary Materials S1 and S2 are submitted with this manuscript. Monthly gap-filled soil moisture layers derived from the approaches proposed in this work can be acquired at Hydroshare, https://bit.ly/31yxfQm, https://www.hydroshare.org/resource/f0091cf90bcc4487bf401ca19783d1eb/.

**Author Contributions:** R.M.L., M.G., and R.V. conceived and designed the research. R.M.L. and M.G. developed the code for processing and analyzing the data. R.M.L. wrote the first draft of the manuscript with input from R.V, M.G. and D.R. All authors contributed to interpretation of the results, reviewed, and approved the manuscript. R.V. and M.T. supervised and coordinated the research team. All authors have read and agreed to the published version of the manuscript.

**Funding:** This study was funded by a University of Delaware Strategic Initiative research grant and the National Science Foundation (OAC grant#1724843).

**Acknowledgments:** We thank Inder Tecuapetla for his input in the conceptual analysis of spatio-temporal correlation and Paula Olaya and Joe Teague for their comments to improve this work. MG acknowledges the Mexican National Council for Science and Technology (CONACyT) for a PhD Fellowship (#382790).

**Conflicts of Interest:** The authors declare no conflict of interest.

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
