# Peer review of "Spatial Gap-Filling of ESA CCI Satellite-Derived Soil Moisture Based on Geostatistical Techniques and Multiple Regression"

_remotesensing, doi:10.3390/rs12040665_

Round 1
Reviewer 1 Report
I do commend the authors for the efforts they have made in following my comments/suggestions. Using the latest version (i.e., v4.5) of the ESA CCI soil moisture dataset sounds more reasonable. Besides, adding the regression kriging (RK) method to the other two already used, made the analysis more robust.
I have only a minor comment, related to use of a reference almost by the same authors of this submitted work (i.e., number 22), when talking about the multiple gaps across the world of the ESA CCI soil moisture product (e.g. lines 63, 73). Please, use an "official" source from the ESA CCI SM Team.
Concerning the same reference, check also the way you cited it.
Line 136, is December 2018, as indicated in Table 1.
Author Response
We appreciate the support from Reviewer 1.
Reference 22 has been replaced by a reference from the ESA CCI team as suggested.
The typo in line 136 has been corrected an it now states the same year as in the linked table “2018”
Additionally, as suggested in the guidelines for authors, data produced in our work has been integrated into a data repository, “Hydroshare”. The URL as well as the assigned DOI have been included In the manuscript, within the supplementary material section.
Reviewer 2 Report
Thanks for clarifying the representativeness issue. I believe it now suitable to be published in Remote Sensing.
Author Response
Thank you
Reviewer 3 Report
As stated from my previous review. I have no further comments. Thanks.
Author Response
Thank you
This manuscript is a resubmission of an earlier submission. The following is a list of the peer review reports and author responses from that submission.
Round 1
Reviewer 1 Report
The paper deals with the comparison of two different geo-statistic approaches for filling spatial gaps of ESA CCI satellite-derived soil moisture dataset (v3.2). The topic is quite interesting, even if, considering the investigated product, there is available a more recent version (v4.4) where the issue related to above-mentioned gaps, has been already taken into account. In the v4.4 version, unreliable retrievals are masked out, producing the gaps the authors several times refer to (e.g., line 70 "this version still contains gaps across many regions of the world"). These gaps, if not present in v3.2, are likely to refer to “wrong” measurements.
In addition to this aspect, the main issue I can see in the paper is that, applying an Ordinary Kriging (OK) interpolation on a geophysical parameter (soil moisture in this case), may help in fill gaps in its spatial distribution over a specific area. I can’t see any novelty in this result, as partially stated by the authors (lines 402-405). Using the GLM method could effectively produces an added-value for the topic, but the quality of the achieved results is lower than OK, indicating that more work have to be carried out.
Reviewer 2 Report
Thank the authors for taking efforts to improve this manuscript. Most of the response and revisions are valid to me except one remained concern that, as the authors sated, collocated stations within the same CCI grid are individually compared to the same pixel to “retain the overall observation-estimation pairs” (L277). I understand the motivation of doing this, but its potential impact is still not well addressed. The response to my previous specific comment#3 and corresponding revisions made in L276-277, however, is not enough regarding this issue. I would suggest the authors to further conduct a small investigation on the NASMD stations to see how many of these collocated stations exist and whether it will actually impact the overall analyzing result (hopefully not), or I believe this should be recognized as another remained problem and thus be incorporated into L441-452.
Reviewer 3 Report
Thanks for carefully revise the manuscript according to all comments and suggestions. I have no major comments but recommend the authors add some recent published papers by ESC CCI soil moisture team and gap filling paper for SMAP/sentinel-1 soil moisture product.